# Uncovering the melt: UAS and in-situ sensor synergies reveal DOC pathways in a northern peatland

Petra Korhonen<sup>1</sup>, Pertti Ala-Aho<sup>1</sup>, Bjørn Kløve<sup>1</sup> & Hannu Marttila<sup>1</sup>

<sup>1</sup>Water, Energy and Environmental Engineering Research Unit, University of Oulu, Oulu, Finland

5 Correspondence to: Petra Korhonen (petra.korhonen@oulu.fi)

Abstract. Spring snowmelt is a critical period for dissolved organic carbon (DOC) export from northern boreal peatlands, yet the spatiotemporal dynamics of this process remain poorly understood. To reveal the spatial patterns, we used a novel combination of high-resolution Unmanned Aircraft System (UAS) snow depth mapping, topographic wetness index, and high-frequency stream monitoring. Our results show that substantial DOC leaching is triggered after widespread snow cover depletion, likely due to thawing of surficial peat layers. High-resolution UAS snow surveys captured the progression of snowmelt from drier, south-facing slopes and forested areas toward wetter fen areas, with the expansion of snow-free areas in high-wetness zones initiating hydrological connectivity and rapid DOC flushing. Event-based hysteresis and flushing analyses enabled by high-frequency stream monitoring revealed transitions from deeper to more surficial flow paths towards the final peak melt. The integration of high-resolution spatial and temporal datasets enabled the detailed identification of DOC transport mechanisms during the snowmelt period. These findings underscore the sensitivity of peatland carbon dynamics to late winter processes and snow conditions, highlighting their potential vulnerability to future shifts in climate.

#### 1 Introduction

Northern peatlands have been recognized as key components in the carbon cycle due to their extensive coverage in northern latitudes (30-90 °N) and long-term capacity for carbon accumulation (Alexandrov et al., 2020; Qiu et al., 2020; Xu et al., 2018). In addition to storing carbon, they are also hotspots for dissolved organic carbon (DOC) export on the catchment scale (Croghan et al., 2024; Laudon et al., 2011; Rosset et al., 2019; Tang et al., 2018) and globally, with 58 % of the global peatland DOC exports originating from boreal regions (Rosset et al., 2022). Aquatic carbon export not only contributes substantially to the peatland net ecosystem carbon balance (Nilsson et al., 2008; Tong et al., 2025) but also has a significant role in downstream water quality and biogeochemical cycling (Kritzberg et al., 2020; Oni et al., 2013), and can lead to increased CO<sub>2</sub> emissions caused by mineralization and evasion downstream (Dinsmore et al., 2010; Serikova et al., 2018). The current concern is that climate change and the associated hydrological shift will alter the peatland carbon balance, although the hydrological responses are complex and remain hard to predict (Qiu et al., 2020; Rosset et al., 2022; Zhang et al., 2022). Contrasting responses may occur in one area or even within one peatland complex (Zhang et al., 2022), highlighting the heterogeneous nature and variable hydrological connectivity at peatlands. Still, the spatiotemporal carbon transport processes are poorly understood (Griffiths et

35

55

al., 2017; Werner et al., 2021), especially during the snowmelt period. Due to strong seasonality in water and carbon cycles, northern peatland ecosystems are sensitive to ongoing climatic change, particularly to snow conditions (Campbell and Laudon, 2019; Croghan et al., 2024; Marttila et al., 2021). Thus, there is an urgent need for documenting and understanding the relevant processes. Particular focus should be given to improved monitoring to facilitate a process understanding during critical periods for ecohydrological connectivity, such as spring snowmelt (Burd et al., 2018; Marttila et al., 2021).

In northern catchments, spring snowmelt dominates annual DOC exports, and studies have found that it contributes 30-64% of annual fluxes (Dyson et al., 2011; Leach et al., 2016; Olefeldt et al., 2013). Climate change is expected to severely modify winter conditions in northern areas, including changes to snow conditions, namely snow accumulation (Pulliainen et al., 2020), duration of snow cover (Luomaranta et al., 2019), and snowmelt timing and rate (Musselman et al., 2017). This will have 40 significant consequences for the catchment hydrological regime, with earlier and less severe spring floods placing more importance on winter streamflow (Teutschbein et al., 2015). As hydrology exerts first-order control on DOC export (Wen et al., 2020), changes in snow conditions and snowmelt timing will directly affect the timing and magnitude of DOC exports (Campbell and Laudon, 2019; Laudon et al., 2013). Changing winter conditions may also impact seasonal soil frost and freezethaw cycles, which can modify hydrologic flow paths and soil DOC concentrations (Ågren et al., 2010; Campbell and Laudon, 2019), affecting the amount of DOC that is available to be transported during snowmelt (Ågren et al., 2012; Campbell et al., 2014). Enhanced soil frost during winter has been found to cause higher and delayed DOC peaks during snowmelt (Ågren et al., 2012; Campbell et al., 2014; Tiwari et al., 2018), and more frequent freeze-thaw cycles can alter soil DOC concentrations (Yu et al., 2011), peat hydraulic properties and water flux (Liu et al., 2022). Alternatively, reduced seasonal soil frost could lead to the earlier activation of subsurface flow paths, increased connectivity, and DOC transport during snowmelt (Croghan et al., 2023; Hinzman et al., 2020; Laudon et al., 2011). These complex trajectories highlight the need to understand how snowmelt impacts DOC transport processes.

Peatlands should be considered as mosaics rather than uniform landscape units, as they are characterised by spatial variability in, for example, hydrological connectivity, groundwater influence, vegetation, and microtopography (Isokangas et al., 2017; Rosset et al., 2020; Vitt et al., 2022). Hydrological connectivity, mainly defined by wetness and water table levels, represents a key factor regulating DOC transport (Knapp et al., 2022; Prijac et al., 2023), which potentially varies, both spatially and temporally, very quickly during snowmelt in peatlands where the water table is naturally high (Isokangas et al., 2017; Laudon et al., 2011; Peralta-Tapia et al., 2015). To assess the spatial variability, topographical wetness index (TWI) has been proposed as a tool to identify hydrological connectivity and active source zones for DOC transport (Knapp et al., 2022; Werner et al., 2021). During snowmelt, activation of sources and carbon transport processes may also be impacted by snow conditions and timing of melt (Campbell and Laudon, 2019; Croghan et al., 2023; Marttila et al., 2021). However, to our knowledge, spatiotemporal variations in snow cover have not yet been combined with TWI to investigate DOC transport during snowmelt.

Snow depth and melt rates can express local or small-scale spatial variation caused by microtopography and vegetation (Litaor et al., 2008; Sannel, 2020; Shirley et al., 2025), and wind processes causing redistribution of snow (Meriö et al., 2023; Shirley et al., 2025). Traditional point-scale snow measurements often fail to capture small-scale spatial variation in snow cover, have limited spatial coverage (tens to hundreds points), and thus can lead to severe over or underestimation, especially during the snowmelt period, as spatial variation in snow depth typically increases towards the end of the snow-covered period (Grünewald et al., 2010; Meriö et al., 2023). Spatial distribution of snow cover can be efficiently monitored with the latest remote sensing technology. However, satellite data is limited by coarse spatial or temporal resolution and issues with cloud cover (Dong, 2018). Thus, novel high-resolution monitoring techniques are needed to assess small-scale snow cover heterogeneity in a resolution relevant to peatland microtopography (







outlets to accurately capture and understand their critical role in landscape carbon exports (Rosset et al., 2019, 2022). In this study, we deployed high-frequency in-situ DOC and hydrological monitoring at small peatland trickle along with UAS mapping during the peak snowmelt period of 2024 at Puukkosuo fen at the Oulanka research station, northeastern Finland. We aimed to i) document spatiotemporal variations in snow cover melt and hydrological connectivity using frequent UAS surveys; ii) link the spatial snowmelt processes to DOC export using high-frequency stream monitoring; and iii) building on the novel datasets, identify the key spatiotemporal hydrological and DOC transport processes occurring during the spring melt period.

#### 2 Methods

#### 2.1 Study area

The research site, Puukkosuo fen, is located in the Oulanka National Park (66.38° N, 29.31° E), northeastern Finland. The mean annual air temperature measured at the Oulanka research station is 0.7°C, and annual precipitation is 559 mm (1995–2024, FMI). The area is characterized by relatively long, snowy winters. The average duration of the snow cover is 194 days, typically ending in May, and the snow depth reaches a maximum of 80 cm on average (1995-2024, FMI). Puukkosuo is a sloping fen peatland located within the north-boreal aapa mire zone. It receives water inputs as groundwater discharge from the hills on the northwestern and northern edges, as well as surface runoff from the upslope catchment area (Figure 1), covering an area of 35 ha and comprising 74% forest and 24% peatland. Waters from Puukkosuo eventually drain to the Oulankajoki River. The fen covers an area of approximately 6 hectares with peat depths ranging from less than 1 meter near the borders and up to 4.6 meters in the center. Due to consistent groundwater discharge, Puukkosuo has a pH close to neutral and provides habitat for several endangered species. Vegetation is mainly open or sparsely tree-covered, dominated by *Sphagnum* mosses, *Carex* sedges, and some Scots pines.

Puukkosuo fen is part of the EcoClimate environmental monitoring programme maintained by the Oulanka research station, University of Oulu, and is equipped with extensive research infrastructure, including high-resolution hydrological and meteorological monitoring and snow measurements. The groundwater level and temperature at the fen were measured in a dip well with a piezoresistive pressure sensor (Decentlab GmbH) measuring at a 10-minute interval. Continuous water level measurements were corrected and converted into water table depth (WTD) using monthly manual water level measurements carried out during the snow-free period as a reference. In addition, air temperature and precipitation were monitored continuously (1-minute interval) at the research site, and precipitation samples were collected at the Oulanka research station after rain or snowfall events for isotope analysis. Precipitation sampling is maintained by the Oulanka research station, approximately 800 meters from the Puukkosuo fen.

Figure 1. Map of the study area and measurement locations at Puukkosuo fen (left figure) and upper catchment boundaries (right figure). Topographic map © NLS 2024.

## 2.2 In-stream monitoring



Water quality was measured continuously (15-minute intervals) at the stream gauging station installed in the peatland outlet, where TriOS OPUS spectral sensor (TriOS GmbH) was installed for water quality measurements. The OPUS sensor measures the full absorption spectrum from 200 to 360 nm, from which spectral analysis is performed to provide readings for the concentration equivalents (eq) of DOC and total suspended solids (TSS). The sensor was cleaned regularly to prevent fouling. Flow measurements (1-minute intervals) were obtained with a 1-inch Parshall flume and UGT502 ultrasonic water level sensor (Ifm electronic GmbH), measuring water level at the crest of the flume and converted it into discharge (Q in L s<sup>-1</sup>) using the equation provided by the manufacturer.





In the same location as in-stream monitoring, weekly grab samples were taken to the laboratory for DOC analyses, which were used to calibrate TriOS OPUS measurements. Samples were first filtered with 1.2 µm GF/C filters (Whatman) and acidified with 2 M HCl before being stored in the dark at 4°C until analysis. DOC concentration was analysed in the Oulanka research station laboratory (Shimadzu TOC-L Analyzer). As laboratory measurements for TSS were not available, the TSS used in this study will refer to TSS equivalent (TSSeq).

In addition, daily water samples were collected for stable water isotope analysis. Samples were collected in 15 ml plastic tubes and stored in the dark at 4°C until analysis. Any samples with visible particles or organic matter were filtered with 0.45  $\mu$ m syringe filters before analysis. The isotopic composition of oxygen ( $\delta^{18}$ O) and hydrogen ( $\delta^{2}$ H) was analyzed at the University of Oulu laboratory with a Picarro 2140-I cavity ring-down spectrometer (CRDS) and were standardized to Vienna Standard Mean Ocean Water. The average measurement precision for all  $\delta^{18}$ O samples used in the study was 0.02‰. Measurement accuracy calculated based on the mean of measured standards was 99.93%.

#### 2.3 UAS surveys and snow measurements

The snow data used in the study was collected through UAS flights and sampling in May 2024. Five UAS campaigns were completed during the peak snowmelt period on 1, 14, 15, 16, and 17 May. In addition, UAS data collected on 31 May was used as a snow-free reference in constructing snow depth maps. All UAS surveys were performed using DJI Mavic 3M, at a 60 m altitude with 70% side and 80% frontal overlaps, resulting in approximately 1.5 cm ground sampling distance for RGB. Mavic 3M was equipped with a Real-Time Kinematic (RTK) GNSS (Global navigation satellite system) module, producing geotagged images with centimeter-level accuracy. In addition, during UAS campaigns, 5–6 ground control points (GCPs) constructed of painted aluminium plates were surveyed with a Emlid Reach RS3 high-precision RTK GNSS receiver for georeferencing. Snow depth at the site was also monitored as a point measurement with a 10-minute interval using a USH-9 ultrasonic snow depth sensor (SOMMER Messtechnik GmbH). In addition, manual snow depth measurements were done during each snow-covered campaign from a snow course transecting the peatland (Figure 1).

During the field campaigns, snowpack samples were collected for stable water isotope analysis. The sample was collected by coring the whole snow profile from the top to the base of the snowpack with a snow tube corer (diameter 3.5 cm). The sample was put into a plastic bag, where it was transferred to 15 ml plastic tubes once completely melted. Also, snow density and snow water equivalent (SWE) were measured at the same sampling location with a snow water equivalent sampler (diameter 10 cm).



## 2.4 Data Analysis

## 2.4.1 UAS data processing

Images taken during each survey were imported and processed in Agisoft Metashape 1.7.3, a processing software employing the Structure-from-Motion (SfM) technique with Bundle Adjustment (BA) method to create high-resolution digital elevation models (DEMs) (Westoby et al., 2012). Processing followed the methods and parameterization described in Ikkala et al. (2022). First, all images with motion blur or low quality were identified and removed. Then, images were aligned to create photogrammetric tie points, generating a sparse point cloud (James et al., 2017), and GCPs were imported as markers and manually verified for all found projections. Although all flights had RTK on board, further georeferencing with one or more GCPs is recommended to correct possible elevation bias (Rauhala et al., 2023). 5–6 GCPs were used, depending on the survey, and for each, two of the GCPs were selected as check points, which are not included in georeferencing but are used to validate the accuracy of the produced model. After georeferencing, poor accuracy tie points were filtered, and camera optimization was performed. Finally, a point cloud was generated with high-quality and moderate depth filtering settings and exported in an LAS format. Additionally, orthomosaics were generated from the point cloud and exported in a TIF format (Figure S1 in the Supplement). Processing information and final model accuracies for each UAS survey, according to Agisoft Metashape, can be found in Table 1.

Table 1. UAS data processing information and ground control point (GCP) root mean square errors (RMSE) for control and check points according to Agisoft Metashape.

| Campaign | Number of aligned images | Ground<br>resolution<br>(cm/pix) | Tie<br>points | RMS<br>reprojection<br>error (pix) | Number of<br>GCPs (check<br>points) | Control points |                | Check points |                |
|----------|--------------------------|----------------------------------|---------------|------------------------------------|-------------------------------------|----------------|----------------|--------------|----------------|
|          |                          |                                  |               |                                    |                                     | XY RMSE (cm)   | Z RMSE<br>(cm) | XY RMSE (cm) | Z RMSE<br>(cm) |
| 1 May    | 429                      | 1.53                             | 152,260       | 1.09                               | 3 (2)                               | 1.77385        | 3.99443        | 2.58502      | 2.05894        |
| 14 May   | 430                      | 1.53                             | 167,451       | 0.591                              | 3 (2)                               | 2.37262        | 3.23791        | 1.75796      | 3.56757        |
| 15 May   | 420                      | 1.53                             | 155,784       | 0.78                               | 3 (2)                               | 2.10955        | 1.26968        | 1.44172      | 1.17555        |
| 16 May   | 423                      | 1.49                             | 144,307       | 0.776                              | 4 (2)                               | 2.55652        | 1.01153        | 2.15228      | 2.06566        |
| 17 May   | 423                      | 1.53                             | 149,091       | 0.808                              | 4 (2)                               | 2.39675        | 0.818499       | 1.67493      | 0.78248        |
| 31 May   | 420                      | 1.53                             | 199,850       | 0.778                              | 3 (2)                               | 3.51992        | 1.71885        | 2.24853      | 2.82244        |


Further processing and data analysis were conducted using R version 4.4.0 in RStudio. First, statistical outlier removal (SOR), together with a cloth simulation filter (CSF), was applied to filter non-ground points from LAS point clouds. CSF is a point cloud filtering method simulating a cloth being dropped on the inverted point cloud surface, and the intersection of the cloth






nodes and corresponding points represents the ground (Zhang et al., 2016). As CSF is sensitive to outlier points, it is vital to use an outlier filtering method before implementing ground classification (Zeybek and Şanlıoğlu, 2019; Zhang et al., 2016). Based on multiple trials, the best parameters selected for SOR were k = 20, m = 5, and for CSF were slope processing = TRUE, rigidness = 1, class threshold = 0.1, and cloth resolution = 1. R Packages LidR (Roussel and Auty, 2016) and RCSF (Roussel and Qi, 2018) were used for point cloud processing. Finally, filtered point clouds were rasterized to 10 cm resolution DEMs with package terra (Hijmans, 2020). Snow depth maps were calculated with a method known as DEMs of Difference (DoD) (Rauhala et al., 2023) by subtracting the snow-covered DEM from the bare ground reference. Snow depth mapping with SfM photogrammetry and DoD method is sensitive to errors caused by environmental factors during the survey, as well as underlying vegetation (Harder et al., 2016). Thus, DoDs (i.e., snow depth maps) were aggregated to a 30 cm cell resolution, reducing the noise and outliers but still maintaining sufficient resolution to identify small-scale variation in snow depth. The accuracy of the produced snow depth maps was estimated by comparing the UAS-produced snow depth to manual snow course measurement points. Differences in manual snow course and UAS snow depth measurements were used to calculate root mean square error (RMSE) metrics for each survey. The accuracy of the manual snow depth measurements was not considered when calculating the error metrics.

Using the UAS-generated snow depth maps, we calculated the proportion of snow-covered area (SCA), mean and median snow depth, and the change in snow depth ( $\Delta SD$ ) between surveys. Preliminary analysis of the snow depth maps revealed that 210 the CSF algorithm struggled to identify the ground and non-ground points, especially in the later snow surveys where part of the ground was already snow-free, thus misclassifying vegetation as ground and causing positive artifacts in snow depth maps. This caused some pixels to have a positive change in snow depth ( $\Delta$ SD) between the surveys. These positive artifacts were interpreted as no change, as they can be assumed to occur in areas where the snow depth had reached zero. All negative artifacts in SD were disregarded in further snow statistic calculations. The estimate of SWE for each raster cell was calculated by 215 multiplying the snow depth by the field-measured snow density. Although manual snow density and SWE were measured as a one-point measurement, the method should provide a sufficient estimate for SWE for the study area since it is fairly homogenous, and variation in snow depth can be expected to override spatial variation in snow density (López-Moreno et al., 2013). The produced SWE maps were used to calculate mean and median SWE, and the total volume of stored water in the snow in the study area.

DEM produced from pre-classified 5 points m<sup>-2</sup> LiDAR data from National Land Survey of Finland (NLS) (2024) was used to delineate the catchment area for Puukkosuo fen and understand the soil moisture and flow path patterns. Topographical wetness indices (TWI) can serve as a proxy for soil moisture and hydrological connectivity (Knapp et al., 2022; Riihimäki et al., 2021). In this study, the DEM of the whole catchment area was used to calculate the Saga wetness index (SWI) (Böhner et al., 2006), which is a modification of the commonly used TWI. SWI is calculated as:




$$SCA_{M} = S_{max} \left(\frac{1}{t}\right)^{\beta \exp(t^{\beta})} \text{ for } S < S_{max} \left(\frac{1}{t}\right)^{\beta \exp(t^{\beta})}$$
 (1)

SWI = 
$$\ln \left( \frac{SCA_M}{tan_{\beta}} \right)$$
 (2)

where SCA is the specific catchment area, t is a suction parameter, and β is the terrain slope angle. The difference between SWI and traditional TWI is that the SWI algorithm uses iterative modification of the multiple flow direction method (Freeman, 1991) for calculating the specific catchment area for each grid cell (Böhner et al., 2006). The t parameter is defined by the user to control the suction effect, i.e., the capillary attraction in the soil voids, simulating the suction or draw of flow from adjacent pixels, which is expected to increase in flatter terrain (Riihimäki et al., 2021; Winzeler et al., 2022). This approach makes SWI more suitable for flat areas such as peatlands (Böhner et al., 2006; Kopecký et al., 2021). Sink filling, calculation of SWI, and flow accumulation raster based on multiple flow direction algorithm were implemented for a 0.5 m resolution DEM in SAGA GIS 7.8.2 software, following the method described in Ikkala et al. (2022). The SWI cells were further classified as low, medium, and high based on quantiles of SWI values to identify changes in snow cover in areas differing in topographically derived wetness conditions. For this purpose, snow depth maps were resampled to a 50 cm resolution to match the SWI raster.

#### 2.4.2 Time-series analysis

Although the Trios OPUS sensor was cleaned automatically and manually, at some time periods, sensor fouling was observed to affect the water quality measurements, resulting in a gradual increase in DOCeq and TSSeq values, followed by a sharp decline after sensor cleaning. We used a systematic method to correct the data values during the fouling periods. The periods of fouling were first identified from the data using the maintenance visit records and then corrected with linear scaling to match the measurement preceding the sensor cleaning to the post-cleaning measurement. A rolling average with a 1-hour window size was then used to smooth out the sensor noise. Following the instructions of the manufacturer, local calibration for scaling DOCeq measurements to DOC mg L<sup>-1</sup> was then created by plotting DOCeq against laboratory-measured weekly DOC grab samples with an offset of 0 (DOC = 0.155 \* DOCeq, n = 37, Figure S2). All in-situ monitoring data was harmonized to hourly averages for further analysis.

Any missing values in Q or DOC, covering 5.9% and 6.3% of the study period for Q and DOC, respectively, were interpolated to produce a continuous time series for further analysis. Baseflow was first separated from the hydrograph by using the Lyne and Hollick method with R package grwat (Samsonov, 2022). Events were defined from the streamflow data as periods when discharge exceeded baseflow by 20% (Blaen et al., 2017; Lloyd et al., 2016) for at least a 10 h duration. The 10 h minimum


duration appeared to provide a suitable threshold for detecting events in the small peatland outlet stream, where the stream hydrograph can be flashy, especially during the spring freshet. If the detected event was double-peaked, it was split into two separate events, ensuring that each event included a rising and falling limb (Lloyd et al., 2016). Event delineation resulted in 14 events identified during the study period, of which two were disregarded due to the absence of distinguishable rising and falling limbs, resulting in unclear loop geometry. For the remaining 12 events, maximum discharge, mean and maximum DOC concentration, total DOC load, and hourly DOC load (DOC load divided by event duration) were calculated.

The hysteresis index (HI) was calculated for each event following the method outlined by Lloyd et al. (2016), where Q and DOC were first normalized for each individual event, and HI was calculated at every 5% of normalized discharge. HI provides a value ranging between -1 and 1, where negative values correspond to anticlockwise hysteretic patterns and positive values to clockwise hysteretic patterns. HI close to 0 indicates a lack of hysteretic relation, while HI approaching -1 or 1 indicates a strong hysteretic relation (Lloyd et al., 2016). The terms clockwise and anticlockwise refer to the direction of the hysteresis loop when concentration is plotted against discharge. Clockwise hysteresis is characterized by the peak concentration occurring during the rising limb of the event, whereas anticlockwise hysteresis occurs when the concentration peak lags the discharge. In the context of DOC transport, clockwise hysteresis is associated with quickly activating flow paths or proximal sources, which may lead to the exhaustion of carbon supply towards the end of the event, while anticlockwise patterns may suggest distal sources of carbon, the activation of DOC transporting flow paths later in the event, and/or different transit times for water and DOC (Lloyd et al., 2016; Vaughan et al., 2017; Williams, 1989).

The flushing index (FI) was calculated to explore the behavior of DOC on the rising limb of each event (Vaughan et al., 2017). FI is defined as the difference between the concentration at the point of peak discharge and the concentration at the beginning of the event, making it equal to the slope of the line. Similar to HI, FI also ranges from -1 to 1, where negative values suggest a decreasing DOC concentration on the rising limb (i.e., a diluting effect) and positive values indicate an increase in concentration (i.e., the flushing of DOC). Positive FI may occur when plentiful proximal sources are transported early during the event and/or from the transport limitation of solute, while diluting can occur when DOC sources are limited and/or quickly depleted (Shatilla et al., 2023; Vaughan et al., 2017).

#### 3 Results



#### 3.1 Spatiotemporal variations in snow depth

Snow cover and melting exhibited spatial variation associated with tree cover, topography, and wetness. On the first survey on 1 May, the highest snow accumulation was observed around the borders of the fen, whereas the central area, areas with higher tree cover, and the south-facing slopes had the lowest snow depth (Figure 2). During the peak melt period, when daily surveys were conducted (14 –17 May), the hillslopes and areas with higher tree cover were the first to become snow-free on


15 May, followed by the central areas of the fen on 16 May. By 17 May, the central area had nearly completely melted, and only residual patches of snow remained, primarily along the boundaries of the fen. Variation in snow depth was greatest during the first survey on 1 May and decreased towards the end of the study period (Figure 3). In addition to natural snow distribution patterns, some anthropogenic influences were also evident in the UAS snow depth maps. Snowmobile tracks crossing the fen are visible on snow depth maps, creating consolidated and slowly melting tracks in the snow. Based on observations on-site, the snowmobile tracks developed into thick ice covers while melting proceeded. Overall, while UAS-derived snow depth maps demonstrated reasonable accuracy, their performance was highly dependent on survey conditions, as expected. The snow depth maps had a consistent negative bias (Mean Error), i.e., they tended to underestimate the snow depth compared to manual snow measurements (Table S1). When considering all surveys, UAS-produced snow depth maps resulted in a mean RMSE of 9.79 cm.

Figure 2. UAS snow depth maps, showing spatial variation in snow depth (cm) for each survey on 1-17 May.




Figure 3. UAS snow depth histograms for each UAS survey on 1-17 May. Vertical lines represent the median snow depth value. Any values below 0 were considered errors and are not shown.

The mean SD declined by 24.81 cm between the first survey on 1 May and the second survey on 14 May, but at that time, over 95% of the area remained snow-covered (Table 2). Considering only the daily surveys between 14 and 17 May, the greatest melting occurred between 14 and 15 May, with a 9.06 cm change in mean SD and 322.29 m³ change in SWE volume. However, the largest change in the snow-covered area occurred between 15 and 16 May, when SCA decreased by 40.71%. Between the last two surveys on 16 and 17 May, greater change occurred in the snow-covered area, which decreased by 17.9%, while changes in snow depth and volume were minor compared to previous days.

Table 2. Snow-covered area (SCA), mean and median snow depth (SD), volume of snow water equivalent (SWE), mean and median SWE for each UAS survey, and manual SD and SWE measurements.

|        | SCA (%) | Mean SD (cm) | Median SD (cm) | SWE (m <sup>3</sup> ) | Mean SWE (mm) | Median<br>SWE (mm) | Mean SD (cm) manual | SWE (mm)<br>manual |
|--------|---------|--------------|----------------|-----------------------|---------------|--------------------|---------------------|--------------------|
| 1 May  | 99.58   | 46.80        | 48.52          | 13207.05              | 193.30        | 200.40             | 57.57               | 237.35             |
| 14 May | 95.71   | 21.99        | 23.23          | 548.91                | 8.36          | 8.83               | 32.14               | 11.46              |
| 15 May | 82.41   | 12.93        | 12.92          | 226.62                | 4.01          | 4.01               | 26.07               | 4.71               |
| 16 May | 41.70   | 6.23         | 4.95           | 71.25                 | 2.49          | 1.98               | 10.71               | 0                  |
| 17 May | 23.80   | 4.66         | 3.22           | 43.36                 | 2.66          | 1.84               | 6.43                | 0                  |

Between 1 May and 14 May, snow depth change ( $\Delta$ SD) (Figure 4) was higher in the most open areas and south-facing slopes compared to north-facing slopes and the northeastern areas with more tree cover. In addition, the snowmobile tracks crossing the fen showed a slower decline in snow depth compared to the surrounding areas. During the daily surveys, spatial variation in  $\Delta$ SD was less evident between 14 and 15 May but increased afterwards. From the daily surveys, spatial variation was highest



between 15 and 16 May, when the largest  $\Delta$ SD occurred in the central areas of the fen and the area adjacent to the natural spring. Between 16 and 17 May, the most substantial changes in SD were observed in areas with the greatest snow depth at the beginning of the study period, particularly near the borders and northern and northeastern parts of the fen.

Figure 4. Maps and boxplots showing spatial variation in snow depth change ( $\Delta SD$  cm) between subsequent UAS surveys on 1-17 May. Scale represents the absolute values, where all changes reflect a decline in snow depth.

When we used the SAGA wetness index (SWI) for the snow-free areas (i.e., areas with SD  $\leq$  0 cm) (Figure 5) to evaluate the interaction between the study site topography and wetness and snowmelt processes, increasing values were noted throughout the surveys, with mean SWI increasing from 7.27 to 9.02 between 1 and 16 May. This reflects melting progressing from drier to more wet areas. This pattern was also seen by differentiating snow depth by each SWI class (Figure 6a), with areas classified as having high SWI exhibiting greater snow depth compared to the low or medium SWI classes until 15 May. However, by 16 and 17 May, snow depth in high SWI zones had decreased to levels comparable to or lower than those in the other classes. The  $\Delta$ SD was slightly higher in the high SWI class compared to the low and medium classes between 1–14 and 15–16 May, while during other surveys, there was only a minor difference between the SWI classes (Figure 6b).

Figure 5. SAGA Wetness index (SWI) for melted areas (snow depth  $\leq 0$  cm). Areas with snow cover are shown as transparent in the map. Vertical lines in histograms represent median values.

Figure 6. Boxplots of UAS snow depth (cm) for each survey (a) and snow depth change ( $\Delta SD$  cm) between subsequent UAS surveys on 1-17 May, and (b) by Saga wetness index (SWI) class (low, medium, high). Outliers or any values below 0 were considered errors and are not shown. In (b), the scale represents the absolute values, where all changes reflect a decline in snow depth.

Between 15–16 May, the largest decline in the snow-covered area occurred within the high SWI class (Table 3), where SCA decreased by 49.97%, while the change in the low and medium classes was 35.44% and 47.44%, respectively. Between 16–17 May, mean SWI of snow-free areas exhibited a slight increase, from 9.02 to 9.08, and the largest change in SCA occurred in the medium SWI class, where SCA decreased by 16.26%. High SWI zones had the most persistent snow cover, and by the last survey on 17 May, 17.82% of high SWI class cells remained snow-covered.

Table 3. Snow-covered area by SWI class for each survey.

|        |         | Snow-covered area (%) |          |  |  |  |  |
|--------|---------|-----------------------|----------|--|--|--|--|
|        | Low SWI | Medium SWI            | High SWI |  |  |  |  |
| 1 May  | 98.38   | 99.55                 | 99.65    |  |  |  |  |
| 14 May | 86.23   | 95.57                 | 96.5     |  |  |  |  |
| 15 May | 60.81   | 77.54                 | 81.46    |  |  |  |  |
| 16 May | 25.37   | 30.1                  | 31.49    |  |  |  |  |
| 17 May | 12.96   | 13.84                 | 17.82    |  |  |  |  |








## 3.2 High frequency dissolved organic carbon (DOC) time series

The earliest snowmelt events in the study period occurred in mid-April 2024 (Figure 7). During this time, two snowmelt peaks were recorded in stream flow, both followed by a rise in air temperature (max temperature 10 °C) and rain events (8.68 mm in total). The next melting event was recorded in early May and was similarly initiated by a rise in air temperature (max temperature 7.8°C), followed by a rain event (5.19 mm). Before the second melting event in May, low air temperatures (mean temperature 0.14°C between 6 and 12 May) restrained the melting. The final large snowmelt peak occurred mid-May when rapid melting during the 5-day period caused snow depth to drop from 40 to 0 cm (Figure 7b). Melting during this period was mainly driven by high air temperatures (mean 7.9°C between 12 and 18 May).

The mean DOC concentration measured at the Puukkosuo outlet stream during the study period was 5.13 mg L<sup>-1</sup>. DOC, and TSS concentrations showed a rapid increase during the first recorded snowmelt pulse, which was also the first identified event (Figure 7a & Figure 7d, and Figure 9a), during which the highest DOC concentration in the study period (6.47 mg L<sup>-1</sup>) was recorded. In the following snowmelt events, increases in DOC concentration were smaller and coupled with discharge, causing DOC to display a staircase-like pattern, with each melting event resulting in a slightly elevated base DOC concentration. At the onset of the final snowmelt peak (events 7–12 in Figure 9a) in mid-May, DOC concentrations first increased simultaneously with discharge, reaching a maximum of 5.5 mg L<sup>-1</sup> on 13 May or event 8, but on the following peak discharge days, a diluting pattern was observed where DOC decreased with increasing discharge, and the lowest concentration of 5.06 mg L<sup>-1</sup> was measured on 15 May (event 10). However, on 16 May (event 11), when the highest discharge (5.24 L s<sup>-1</sup>) in the study period was recorded, DOC concentration increased without a clear diluting pattern, reaching 5.66 mg L<sup>-1</sup> until the discharge increased again on the next day (event 12). DOC load also increased towards the end of the snowmelt period, with the highest load (max 103.66 g h<sup>-1</sup>) observed at the last day of the final snowmelt peak (event 12) (Figure 7d). TSS concentration remained more stable before the last snowmelt peak, when the concentrations showed a simultaneous increase with discharge.

Stream  $\delta^{18}$ O values in the study period varied between -17.22% and -13.89% (Figure 7c). The mean  $\delta^{18}$ O value for precipitation during the study period was -12.48%, with more enriched values occurring during warm periods, i.e., when precipitation occurred as rain, not snow. Snowpack  $\delta^{18}$ O values were generally more depleted when compared to stream and precipitation values, ranging from -19.39% to -16.70%. Each snowmelt pulse was displayed as a more depleted  $\delta^{18}$ O in stream water, verifying the presence of snowmelt water in the stream. The most depleted stream  $\delta^{18}$ O values were recorded during the last snowmelt peak between 14 May and 17 May, when the mean  $\delta^{18}$ O was -16.48%. After the snowmelt was complete, stream  $\delta^{18}$ O became less depleted and stabilized quickly, resembling the values before the initial snowmelt pulse in early April.

Peatland groundwater level response in the GW well was very synchronized with streamflow and precipitation until the snowmelt was exhausted (Figure 7e). WTD ranged from -4.5 cm to 5.4 cm, but the water table remained above ground level for most of the study period. Based on the field observations, the GW well was frozen during the study period; thus, high WTD is likely due to meltwater accumulating on top of the ice layer. WTD started to decrease during the final snowmelt peak, although GW temperatures did not begin to rise until a week later.


Figure 7. Water quality and hydrological parameters monitored during the study period April-May. Discharge (Q) and DOC concentration monitored at the stream gauging station (a), air temperature and point snow depth measurement in Puukkosuo (b), precipitation monitored in Puukkosuo and  $\delta^{18}$ O isotope samples of the stream, precipitation, and snow (c), DOC load and TSSeq concentration (d), and water table depth (WTD) and water temperature monitored in the GW well in Puukkosuo (e). Dashed lines represent the dates of UAS surveys.

The cumulative DOC load and discharge during the study period exhibited a similar staircase pattern, with each melting event causing an increase in both DOC load and discharge (Figure 8a). The highest increase in both was observed during the final melt peak, when DOC load increased in parallel with discharge. After the snowmelt was completed, DOC load showed a slightly more increase relative to discharge. The cumulative SWE volume (Table 2) loss was used to estimate water flux from melting snowpack. A large decrease in SWE volume between 1 and 14 May occurred due to the gap in UAS surveys. However, cumulative loss in SWE volume between 14-17 May declined towards the end of the melt peak, while DOC load and discharge increased steeply until the very end of snowmelt. Expansion of snow-free area showed more similar curve with the cumulative DOC load and discharge, with very fast progression during the final melt peak (Figure 8b).

Figure 8. Cumulative DOC load (g), cumulative discharge (m³) during the study period, and cumulative snow water equivalent (SWE) loss (m³) (a) and development of snow-free area (%) (b) derived from UAS surveys. Colors represent the identified events 1–12.

## 410 3.3 Event analysis



Based on the event detection criteria, 12 events were identified during the study period, of which three occurred during the initial melting in early April, three in early May, and the last six during the final melt peak in mid-May (Table 4, Figure 9a). The average event duration was 23 hours. The longest event, number 6, lasted for 60 hours, while event number 11, occurring on 16 May during the peak melt period, was the shortest with a 10-hour duration. The highest mean DOC concentration was measured in event 12, while event 1 had the highest maximum DOC concentration. However, differences in DOC concentrations between events were small, and greater disparities were observed in DOC loads. When considering event duration, event 12, occurring on 17 May, contributed the largest hourly DOC load, while event 1 had the lowest hourly DOC load. The highest total DOC exports occurred in the longest event, event 6, followed by event 12, while event 4 contributed the lowest DOC exports.

Table 4. Timing, duration, maximum discharge, hysteresis index (HI), flushing index (FI), maximum DOC concentration, mean DOC concentration, hourly DOC load, and total DOC load for events identified during the study period.

| Event<br>ID | Start                  | End                    | Duration<br>h | Max Q L | НІ    | FI   | Max<br>DOC<br>mg L <sup>-1</sup> | Mean<br>DOC<br>mg L <sup>-1</sup> | Load g<br>DOC<br>h <sup>-1</sup> | Total<br>load g<br>DOC |
|-------------|------------------------|------------------------|---------------|---------|-------|------|----------------------------------|-----------------------------------|----------------------------------|------------------------|
| 1           | 2024-04-10<br>18:00:00 | 2024-04-12<br>17:00:00 | 47            | 0.69    | 0.12  | 0.40 | 6.47                             | 5.28                              | 8.44                             | 396.82                 |
| 2           | 2024-04-14<br>13:00:00 | 2024-04-15<br>05:00:00 | 16            | 0.71    | -0.64 | 0.47 | 5.05                             | 5.01                              | 12.29                            | 196.69                 |
| 3           | 2024-04-15<br>05:00:00 | 2024-04-16<br>13:00:00 | 32            | 0.86    | -0.29 | 0.48 | 5.12                             | 5.08                              | 14.17                            | 453.39                 |
| 4           | 2024-05-01<br>17:00:00 | 2024-05-02<br>07:00:00 | 14            | 0.45    | -0.65 | 0.59 | 5.12                             | 5.08                              | 8.64                             | 120.95                 |
| 5           | 2024-05-02<br>14:00:00 | 2024-05-03<br>10:00:00 | 20            | 0.74    | -0.78 | 0.80 | 5.28                             | 5.24                              | 13.44                            | 268.90                 |
| 6           | 2024-05-03<br>13:00:00 | 2024-05-06<br>01:00:00 | 60            | 1.39    | -0.24 | 0.45 | 5.30                             | 5.26                              | 19.40                            | 1164.11                |
| 7           | 2024-05-12<br>14:00:00 | 2024-05-13<br>08:00:00 | 18            | 1.09    | -0.84 | 0.48 | 5.29                             | 5.24                              | 19.46                            | 350.25                 |
| 8           | 2024-05-13<br>08:00:00 | 2024-05-14<br>04:00:00 | 20            | 1.80    | 0.03  | 0.76 | 5.50                             | 5.29                              | 30.39                            | 607.80                 |

| 9  | 2024-05-14<br>11:00:00 | 2024-05-15<br>03:00:00 | 16 | 2.67 | 0.07  | -1.00 | 5.26 | 5.23 | 44.49 | 711.90 |
|----|------------------------|------------------------|----|------|-------|-------|------|------|-------|--------|
| 10 | 2024-05-15<br>11:00:00 | 2024-05-16<br>00:00:00 | 13 | 4.92 | -0.55 | -0.44 | 5.21 | 5.12 | 64.59 | 839.68 |
| 11 | 2024-05-16<br>12:00:00 | 2024-05-16<br>22:00:00 | 10 | 5.24 | -0.37 | 0.47  | 5.43 | 5.38 | 78.06 | 780.59 |
| 12 | 2024-05-17<br>09:00:00 | 2024-05-17<br>21:00:00 | 12 | 5.22 | 0.60  | -0.43 | 5.66 | 5.47 | 80.62 | 967.46 |



Events 4, 9, 10, 11, and 12 had UAS snow cover data collected on the same day as the onset of the events. In these events, hourly DOC load and maximum discharge increased together as snow depth declined toward the end of the snowmelt period, and changes in mean snow depth in different wetness zones decreased. Between the final two events, when there was only a minor decrease in mean snow depth, discharge did not show further increase, but DOC load still slightly increased. Snow-covered area showed a slightly different pattern, as DOC load and discharge first increased at a faster rate relative to SCA decline. After event 10, SCA decreased rapidly in all wetness zones with hourly DOC loads still increasing (Figure S3).

Normalized hysteresis loops (Figure S4) exhibited complex behavior, with the majority being anticlockwise, two clockwise,

— or H


and three figure-of-eight or otherwise complex-shaped loops. HI ranged from -0.8 in event 7 to 0.6 in event 12. Mean HI was -0.3 and was negative for most of the events, reflecting a dominant pattern of DOC concentrations peaking on the falling limb of the hydrograph. Event 12 was the only one to show a clear clockwise hysteretic pattern. Events 8 and 9 had weak positive HI very close to 0, indicating a lack of hysteretic pattern for these events. Mean FI was 0.25, ranging from -1 to 0.8. FI was mainly positive but underwent a shift to negative in events 9, 10, and 12, indicating a transition from flushing of DOC during the rising limb to a diluting pattern during the events at the peak melt period with increasing daily maximum discharges (Figure 9a). A quadrant plot of HI versus FI (Figure 9b) revealed that most of the events represented an anticlockwise hysteresis together with flushing behavior, clustering in the bottom-left corner. However, events during the final melt period in mid-May exhibited distinctive flushing and hysteresis patterns, shifting from flushing to diluting and from anticlockwise to clockwise between subsequent events.

Figure 9. Events 1–12 identified during the study period based on stream discharge (a), quadrant plot of hysteresis index (HI) and flushing index (FI) with numbers representing the event ID (b), and a scatterplot of TSSeq and DOC concentrations with point colors representing the event ID (c).

The Spearman correlation was used to identify potential links between DOC and TSS exported during events (Figure 9c). DOC and TSS concentrations during events showed a positive correlation (cor=0.67, p 

During the last two surveys, large snow-free areas first developed in the central parts of the fen, before continuing towards the borders, where the most snow had accumulated. Higher snow accumulation in these areas can be explained by wind distribution processes relocating snow from the open peatland area towards the sheltered edges (Harder et al., 2020; Meriö et al., 2023). Our observations are supported by the findings of Grünewald et al. (2010), who concluded that spatial variation in maximum snow accumulation has a greater impact on snow depletion than variability in snow melt rates. The development of snow-free patches and the subsequent transport of melt energy to adjacent areas are the main controlling factors of snow melt spatial variability (Grünewald et al., 2010). This pattern could be identified in our study area as well, as after the large snow-free areas formed on 16 May, differences in snow depth and snow depth change between the SWI classes decreased, reflecting more uniform melting. However, hydrology had some influence on snow accumulation and/or melting in our study area, as areas with higher wetness identified by SWI classes seemed to have more persistent snow cover and higher snow depths until the very end of the snowmelt. This can be explained by the formation of ice cover in the low-lying areas with high wetness, as observed in the field. With the UAS snow depth mapping technique, this surficial ice cover is captured as snow (Rauhala et al., 2023). More persistent snow cover of high SWI zones can also be explained by topographical depressions accumulating more snow due to wind-driven snow trapping (Shirley et al., 2025), which could potentially further increase the wetness. In addition, the area immediate adjacent to the spring (Figure 2) became snow-free earlier in comparison to the surrounding areas, likely due to warmer groundwater enhancing the melt (Van Huizen et al., 2022). However, the impact is very small-scale, and the physical processes described above are likely the main contributors to spatial melting patterns.





470

475

480

The peak melting documented through UAS surveys proceeded rapidly due to high daily temperatures (mean 7.92°C) and was potentially further accelerated by rainfall (1.97 mm) on the evening of 15 May (Figure 7). During the first days, substantial changes occurred in snow depth and volume, but greater decreases in snow-covered area (SCA) occurred during the last two surveys, particularly on 16 May, when the first large uniform snow-free areas formed and only 41.7% of the area remained snow-covered (Table 2). Substantial losses were also observed in SWE volume, and the highest discharge was recorded on 16 May. On 17 May, SCA still decreased by 17.9%, but changes in SWE volume were minor compared to previous days. However, maximum discharge was still high, which might be explained by more runoff from melting snowpack originating from the upper forested catchment area, as these typically melt later compared to open peatland (Meriö et al., 2023). Another possible explanation is that runoff was routed through flow paths more hydrologically connected to the stream gauging station. The results demonstrated that the decline in snow depth or SWE volume and SCA followed different patterns. Snow depth or SWE volume decreased more gradually, with a rate of decline slowing towards the end of the peak melt period, while SCA showed a sharp decline at the end of the study period. The interplay of these parameters has important implications, as snow mass defines how much snowmelt runoff can be produced, but the absence or presence of snow cover shapes soil thermal regime, soil moisture and flow paths (Dong, 2018; Webb et al., 2018). UAS-based snow monitoring approach has the potential to address the coverage and resolution issues faced with point snow measurements and satellite data and improve monitoring of snow cover at local and catchment scales during the period of rapid snowmelt.







Our second objective was to link the observed spatial snowmelt processes to DOC export with the high frequency stream monitoring. Decreases in snow-covered area and exposure of the peat surface, particularly in high wetness areas, coincided with increasing DOC concentrations and load, which can be associated with increased hydrological connectivity activating DOC-transporting source areas and soil layers (Knapp et al., 2022; Prijac et al., 2023). During the last two surveys, when snow cover distribution had become more uniform between different wetness zones, maximum discharge did not show further increase, but DOC loads still slightly increased. This indicates that once hydrological connectivity was established, further DOC export was governed by source activation rather than by runoff magnitude. Subsurface hydrological connectivity during snowmelt is restricted by soil frost or ice layers at the top of the peat, forcing water to be transported as overland flow or as deeper preferential flow paths (Laudon et al., 2007; Peralta-Tapia et al., 2015). This process can lead to a dilution effect in high flows, as the streamflow consists of snowmelt water with low DOC concentrations (Eskelinen et al., 2016; Leach et al., 2016). This pattern was also observed in our study on 14, 15, and 17 May, with high daily maximum discharge (Figure 7). However, when the soil begins to thaw and meltwater is able to infiltrate, new flow paths and pools of DOC can activate for transport (Croghan et al., 2023; Rose et al., 2023). Thus, depletion of snow cover likely facilitated DOC transport by allowing the thawing of surficial peat layers to begin as the insulating snow cover was removed. This would allow the shift of the dominant flow path to shallow subsurface layers, thus leading to rapid flushing of DOC from peat layers (Birkel et al., 2017). This was supported by the simultaneous decrease in WTD once large snow-free areas were formed, indicating soil thawing. Typically, an increase in water table is presumed to reflect hydrological connectivity (Knapp et al., 2022; Prijac et al., 2023; Rosset et al., 2020), but in Puukkosuo, the water table remains high and fluctuations are small, and we expect the soil frost to be the main factor restricting subsurface hydrological connectivity during spring snowmelt.

The depletion of snow cover and subsequent thawing of peat layers could also boost biological activity and DOC production due to warming soil temperatures (Campbell and Laudon, 2019; Wen et al., 2020). However, production rates of DOC during cold periods are typically restricted by low temperatures, and it is unlikely that the production rate during spring snowmelt could compensate for DOC exports (Campbell et al., 2014; Wen et al., 2020), often resulting in source limitation (Gómez-Gener et al., 2021). Thus, prevailing conditions during the antecedent winter and the previous year's DOC exports largely define the amount of DOC accumulated and available for transport during snowmelt (Ågren et al., 2010, 2012; Tiwari et al., 2018). High DOC export during the preceding summer and autumn can decrease the DOC pool available for transport during the next spring unless production is sufficient to compensate (Ågren et al., 2010). Cold autumns and winters, associated with severe soil freezing, have been found to cause higher DOC peaks during snowmelt (Tiwari et al., 2018), as severe soil freezing and recurring freeze-thaw cycles during winter can increase peat DOC concentrations and lead to delayed leaching of more labile DOC due to physical disruption of soil aggregates (Campbell et al., 2014; Liu et al., 2022; Yu et al., 2011). However, it is the timing and degree of lateral hydrological connectivity that eventually defines when and whether DOC is transported (Knapp et al., 2022; Prijac et al., 2023; Wen et al., 2020).





# 4.2 Peatland shows early activation and rapid hydrological response to melting events

The measured DOC concentrations (mean 5.13 mg L<sup>-1</sup>) in Puukkosuo were overall slightly lower compared to other studies carried out in boreal peatland catchments during snowmelt (Dyson et al., 2011; Gómez-Gener et al., 2021; Leach et al., 2016) but similar to concentrations measured in the subarctic Pallas catchment (Croghan et al., 2023, 2024). DOC concentration in the peatland outlet stream was highly influenced by the onset of snowmelt and the related increases in snowmelt water inputs. Initially, DOC concentration and discharge were low, but the first snowmelt pulse facilitated a peak in DOC and TSS, which was considerably higher than the events occurring during the rest of the study period (Figure 7). A similarly high initial pulse in turbidity was observed by Croghan et al. (2023), who suggested it reflected DOC and particulate organic carbon (POC) mobilization alongside sediment. In our study, DOC was positively correlated with TSS for most of the events, which also suggests the simultaneous mobilization of DOC and particulate matter (Figure 9c). The high concentration peak can be explained by the initial snowmelt pulse enabling reconnection of DOC-rich flow paths and snowmelt water flushing DOC and particulate material stored in the peat matrix (Eskelinen et al., 2016; Prijac et al., 2023). The first event also exhibited a complex figure-eight-shaped loop, indicating heterogeneous flow paths or DOC sources (Rose et al., 2023). After the first event, increases in DOC with discharge were more subtle, but sources remained sufficient to increase DOC concurrently with discharge until the final melting peak, when the first diluting patterns associated with high discharge were observed. Increasing DOC throughout the study period suggests the increasing activation of flow paths and transport of DOC as melting proceeds (Marttila et al., 2021).

Differences in events' DOC load were greater compared to DOC concentration, and most DOC was exported in events occurring during the final melting peak with the highest daily discharges (Table 4), highlighting the coupling of discharge and DOC export. This was also seen as a clear response in cumulative DOC load and discharge to the melting events, and more constant DOC export between the events, resulting in an almost linear increase (Figure 8a). It is possible that part of the exported DOC originated from the upper catchment, especially at the end of the snowmelt period (Ågren et al., 2008, 2012; Laudon et al., 2011; Tang et al., 2018). After snowmelt was completed, cumulative DOC load increased slightly more relative to discharge, indicating DOC production or activation of distal sources as the progression of snow melt enabled hydrological connection to extend within the upper catchment (Croghan et al., 2023). Still, peatland can be expected to have a significant contribution to DOC exports, given that 24% of the catchment area is covered by peatland and the stream gauging station is located immediately at the fen outlet, forcing all incoming water to pass through it (Laudon et al., 2011; Rosset et al., 2019).

The proportion of new event water during snowmelt can contribute over 50% of streamflow in peatland-dominated catchments (Laudon et al., 2007; Noor et al., 2023). The role of snowmelt as a dominant water source is also highlighted in Puukkosuo, as every snowmelt pulse resulted in depleted stream  $\delta^{18}$ O, and during the peak melt period, stream  $\delta^{18}$ O values closely matched those of the snowpack (Figure 7c). This corresponds well with the major decreases in SWE volume observed between 14–16






May, suggesting that a major fraction of the stream water originated from melting snowpack. However, on 17 May, distinctions in stream δ<sup>18</sup>O compared to snow was observed, indicating meltwater infiltration and mixing with soil water as the soil began to thaw and become hydrologically connected (Eskelinen et al., 2016; Muhic et al., 2023). By this time, 76.2 % of the fen was already free of snow cover, and changes in snow depth and volume were minor, explaining the increasing infiltration and decreasing snowmelt signal inferred from stream δ<sup>18</sup>O. After the melt was completed, stream δ<sup>18</sup>O became more enriched and stabilized rather quickly to the initial enriched values closer to local groundwater (not shown here), as also seen in the study by Noor et al. (2023). The rapid response and recovery from snow melt water introduced δ<sup>18</sup>O depletion implies minimal dynamic snowmelt water storage in the catchment, resulting in the active formation of surface runoff and the short duration of snowmelt water presence (Marttila et al., 2021; Peralta-Tapia et al., 2015). The rapid hydrological response is further highlighted by the nearly synchronous responses in streamflow, water quality, and WTD to melting events.

## 4.3 Event analysis reveals DOC transport dynamics and flow path activation during snowmelt

Analysing events' concentration-discharge responses enabled identifying the relevant DOC transport processes. Most of the identified events during the study period exhibited anticlockwise hysteresis together with positive flushing indexes (FI) (Figure 9b), which has been attributed to rapid flushing of DOC and reconnection of carbon-rich subsurface flow paths later in the event in peatland ecosystems, causing concentrations to peak in the falling limb of the event (Prijac et al., 2023; Tunaley et al., 2016). Previous studies in peatland catchments have found both anticlockwise (Burd et al., 2018; Prijac et al., 2023; Tunaley et al., 2016) and clockwise (Ågren et al., 2008; Dyson et al., 2011) hysteretic patterns, and often a combination of both in spring (Croghan et al., 2023, 2024; Rose et al., 2023), highlighting the complex transport dynamics and sources during the snowmelt season (Shatilla et al., 2023).

We observed a change in hysteretic behavior during the final melting peak contrasting the dominant anticlockwise pattern, as the hysteresis index (HI) shifted first to slightly positive, then back to negative, and reached the highest value during the last detected event. A similar shift from anticlockwise to slightly clockwise hysteresis during snowmelt was observed by Croghan et al. (2023), who attributed it to a change from slower acting subsurface flow paths to the activation of dominant DOC transporting pathways or the depletion of DOC sources in the catchment. We also found changing flushing behavior during the last snowmelt peak. A shift to negative FI was observed in events 9 and 10 when daily discharge increased, which can be attributed to a diluting effect of snowmelt water when DOC sources are inadequate to keep up with increasing runoff (Gómez-Gener et al., 2021). However, in event 11, positive FI was again observed despite the highest maximum discharge measured during the day, suggesting the activation of new carbon sources and/or hydrological connections (Vaughan et al., 2017). Event 11 occurred concurrently with the greatest depletion of SCA and snow depth observed in high wetness areas, which can be linked to the expansion of hydrologically connected areas and the associated increase in DOC transport (Knapp et al., 2022). The activation of these hydrological pathways likely primed the rapid transport of DOC for the following day and event 12

(Croghan et al., 2023), when clockwise hysteresis was observed and DOC peaked on the rising limb of the event. However, the return to negative FI in the final event, event 12, might have resulted from the quick depletion of the newly activated sources and/or high water table facilitating overland flow (Croghan et al., 2023; Leach et al., 2016). In fens, upper peat layers can be flushed rapidly, causing a sudden depletion of DOC in pore water (Rosset et al., 2020). Surficial peat layers might also have a limited supply of DOC due to the freeze-out effect (Ågren et al., 2012), which might have resulted in the source limitation and diluting pattern observed in event 12.





Previously, it has been found that most of the exported DOC during spring originates from deeper flow paths, but shifts to surficial carbon pools as melting proceeds (Leach et al., 2016; Rose et al., 2023). The activation of new DOC flow paths has been associated with the completion of snowmelt in the first parts of the catchment, typically being the peatland areas (Croghan et al., 2023; Meriö et al., 2023). Our event analysis results also suggest a change in dominant sources and flow paths, as early melting events exhibited hysteretic and flushing patterns typical for lateral subsurface flow but shifted to more surficial flow path behavior during peak melting (Croghan et al., 2023; Prijac et al., 2023). However, shifting hysteretic and flushing behavior during the final snowmelt peak suggests that the change of dominant flow paths can be fast, and sources are not straightforward. This implies highly dynamic melting processes and flow path development (Rose et al., 2023). Documenting the snowmelt with the UAS surveys suggests that this can be associated with snow cover depletion in the wettest zones, first causing the activation of new surficial DOC flow paths and leading to quick flushing due to increased connectivity. Integrated highresolution stream monitoring and spatial snow cover mapping allowed linking the observed DOC dynamics to the snowmeltdriven DOC transport processes (Figure 10). The role of catchment conditions and spatial variation in hydrological connectivity has often been overlooked in concentration-discharge analysis, and hysteretic relations have been expected to represent the activation of different source areas (Knapp et al., 2022). The integration of novel monitoring technologies holds potential to bridge this gap by explicitly linking spatial and temporal patterns. In addition, further information on variation in pore water DOC concentrations could provide a better assessment of spatial variability, mobilization, and transport processes (Knapp et al., 2022; Vitt et al., 2022).

- → Anticlockwise hysteresis, and flushing of DOC from carbonrich peat layers
- → Low hourly DOC load
- 2 Initial melting occurs in the south-facing slopes and areas with higher tree cover
  - → Increasing water inputs and surficial flow paths increase hourly DOC load and dilution
  - → Slight clockwise hysteresis, or no clear hysteretic pattern

- 3 Depletion of snow cover allows thawing of peat layers and activates new sources and flow paths
  - → Return to anticlockwise hysteresis
- Continuing rapid melting, particularly in high-wetness areas, initiates hydrological connectivity and DOC transport
  - → Flushing of DOC from recently activated areas and peat layers

At the final stage of melting, the main surface water flow paths are activated

- Clockwise hysteresis and dilution of DOC due to depletion of sources and/or fast overland flow
- → Highest hourly DOC load

Figure 10. Conceptual diagram summarizing the observed and hypothesized snowmelt-driven DOC transport processes during the snow-covered and final melting period in Puukkosuo fen. Blue arrows represent the main flow paths contributing to DOC transport.

## **5 Conclusion**


This study used a novel combination of high-resolution UAS snow depth mapping, topographical wetness index, and high-frequency stream monitoring to reveal spatiotemporal snowmelt dynamics, hydrological connectivity, and DOC transport in the northern Puukkosuo fen. By coupling spatial and temporal high-resolution monitoring and multiple analytical techniques, we found that snow cover depletion, particularly in high wetness areas, triggers hydrological connectivity and DOC transport. Event analysis also suggested the activation of new sources over time, likely due to the thawing of surficial peat layers. Overall, shifting hysteresis and flushing behavior were observed during the study period, reflecting changes in dominant flow paths from deeper sources early in the melt to more surficial contributions as the snowmelt period progressed and snow cover started to deplete.

The combination of high-resolution temporal stream monitoring and spatial snow cover mapping allowed detailed insights into the connection between snowmelt progress and stream Q-DOC response. As shown by our results, these processes exhibit rapid variation even on a daily timescale, emphasizing the need for high-resolution monitoring to adequately capture the






processes in play and estimate DOC export during snowmelt. UAS snow monitoring proved to be a cost-efficient method to acquire spatially and temporally resolved information on snow cover changes during the snowmelt period. UAS-based information on snow depth, volume and snow-covered area could also provide more explicit information on these parameters valuable for ecohydrological and biogeochemical modelling and its development. To our knowledge, this study is the first to use UAS surveys to document and link spatial variation in snowmelt-induced hydrological activation to event Q-DOC characteristics. A similar UAS-based monitoring method could provide insights into spatial variation in hydrological connectivity and provide a more robust link between the catchment conditions and events C-Q dynamics during other seasons as well. Introducing spatial mapping is needed to address the spatial hydrological processes associated with DOC transport within peatlands, and we encourage further studies to combine spatially resolved information on catchment hydrological conditions with event C-Q analysis to provide more robust interpretations on activation and contribution of different source areas. To further improve the understanding of the dynamic nature of peatland ecosystems, more information on how different parts of peatland contribute to DOC transport and production is needed.

Our identified processes can be expected to be relevant also in other fen ecosystems within northern boreal regions with comparable seasonal and snowmelt patterns. Our results indicate that hydrological activation is sensitive to late winter processes, as the fen is already activated in the early phase of melting. Thus, earlier or more rapid snowmelt and warmer periods in winter can directly impact hydrological activation and DOC leaching. Also, DOC transport processes may be impacted by any changes in snow, soil frost, and thawing conditions, thus Puukkosuo and similar fens in boreal and arctic regions are likely to be sensitive to climate change impacts. These findings emphasize the importance of snowmelt timing and spatial patterns in controlling peatland DOC export and suggests that water-carbon dynamics may undergo substantial changes under future climate conditions.

## Data availability

Multivariate time series dataset and snow depth maps used in this study are available at https://doi.org/10.23729/fd-15386b07-324e-3a47-86b1-e1a3a9845dba under Creative Commons Attribution 4.0 International (CC BY 4.0) license. Data is published under embargo until 31 March 2026, or until the paper is accepted for publication. During manuscript review the datafiles are available from the corresponding author upon request. LiDAR data was obtained from the National Land Survey's (NLS) Laser scanning data 5 p dataset in 2024.

#### **Author contribution**

Conceptualization: PK, HM, and PAA; Formal analysis: PK; Funding acquisition: HM, PAA, and BK; Investigation: PK; Resources: HM, PAA, and BK; Writing – original draft: PK; Writing – review and editing: PK, HM, PAA, and BK.

## **Competing interests**

The authors declare that they have no conflict of interest.

## Acknowledgements

We thank the Oulanka research station staff, especially Vesa-Matti Kleemola, Juho Lämsä, Eero Koskinen, and Teppo Salmirinne, for their efforts in setting up and maintaining the stream monitoring and sampling. We also thank Karoliina Särkelä for helping with the UAS surveys and Maisha Ahmed for assistance in preparing the isotope samples. Puukkosuo is part of the EcoClimate experimental platform at Oulanka research station. We acknowledge the assistance of AI in creating the R scripts and text editing.

### Financial support

The study has been supported by the Research Council of Finland project Carbon-water interactions in a changing Arctic catchment (347663) and by the European Union — NextGenerationEU RRF project Green-Digi-Basin (347704). Writing was supported by Digital Waters (DIWA) flagship funded by Research Council of Finland, and infrastructure by the European Union — NextGenerationEU RRF project HYDRO-RI-Platform (346163). PAA was supported by the Research Council of Finland project SNOMLT (347348). Snow monitoring was supported by the European Union — NextGenerationEU RRF project CRYO-RI (352758).

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
