# Peer review of "Uncovering the melt: UAS and in-situ sensor synergies reveal DOC pathways in a northern peatland"

_EGUsphere, 2025_

## Author Comment (AC1)

**General comments:**

**RC1C1:** This is an interesting setup touching a weak spot in hydrology and solute mobilization: While connectivity is used to explain solute export dynamics, it is hard to actually map and measure. This study can go a step in this direction by a drone-based assessment of snow cover changes during a major snowmelt event combined with concentration dynamics in the receiving stream. This topic is of great interest to readers of HESS.

While the data and effort are impressive, I am not totally convinced about the way results are described and interpreted. From my point of view the result section can be more concise and the discussion section much more integrative. I see a tendency to overinterpret patterns. My suggestion is a careful revision following the points raised below. I moreover encourage the authors to look for a measure of connectivity in the snow cover data that can be used as an explanatory variable in the event C-Q analysis. Just fraction of area covered by snow or snow depths is not telling a story of connectivity of source zones to the stream. However, measures of spatial connectivity of snow-free area changing over time may provide that.

**RC1A1:** We thank the reviewer for the detailed comments and feedback, and the manuscript will be carefully revised following the comments. We fully agree that hydrological connectivity is a key concept for interpreting solute export dynamics, and that simple metrics such as total snow-free area or mean snow depth alone do not adequately describe connectivity of source areas to the stream.

Motivated by this comment, we explored whether the UAS data could be used to derive a connectivity-relevant descriptor from the spatial pattern of snowmelt. While snow cover limits direct observation of surface wetness or flow paths during spring, the progressive exposure of the peat surface provides information on where potential source areas become hydrologically active. We therefore use the SAGA Wetness Index (SWI), calculated from a snow-free DEM, to characterize the potential hydrological connectivity of those areas as they become snow-free.

Specifically, we calculate the mean SWI of snow-free pixels for each UAS survey as a first-order proxy for the wetness state and potential connectivity of activated areas. We emphasize that SWI is a static, topography-based index and does not describe connectivity dynamically; rather, connectivity emerges through the interaction between static topographic controls and the temporally evolving snow cover.

Despite the limited number of surveys (five during rapid snowmelt), we find that mean SWI of snow-free areas is significantly positively correlated with event maximum discharge and hourly DOC load, and negatively correlated with the flushing index. These

relationships suggest that not only the extent but also the topographic position of snow-free areas influences event-scale hydrological and biogeochemical responses. This additional analysis and a new figure will be included in the revised manuscript, with appropriate caution regarding its exploratory nature.

**Specific comments:**

**Title**

**RC1C2:** For me UAS is not self-explaining and I would even avoid DOC as an abbreviation in the title.

**RC1A2:** Abbreviations in the title will be written in full in the revised manuscript.

**Abstract**

**RC1C3:** L9: I miss a reference to the location and size of the catchment/area to help the reader understanding what dimension you are talking about.

**RC1C3:** Reference to the location and size of the catchment will be included in the revised version.

**Introduction**

**RC1C4:** The introduction reads very well – not much to revise from my point of view. However, I found the isotope analysis as not at all motivated in the introduction. You should mention at some point here and / or in the methods what this analysis is used for.

**RC1A4:** The isotopes are used as supporting data for interpretations on the flow paths and snowmelt inputs. The rationale will be included in the introduction or methods for the revised manuscript.

**RC1C5:** L56: "water table" or better "water levels" is enough.

**RC1A5:** Will be changed in the manuscript.

**RC1C6:** L60: These studies use the TWI at catchment scale or in riparian zones but not in peatlands. Are there also examples from peatlands pointing a dominant topographic control of discharge generation?

**RC1A6:** We agree that flow paths in peatlands are complex and not solely controlled by surface topography, particularly under dry conditions or when vertical gradients dominate. However, during snowmelt and other wet periods when the water table is close to or above the peat surface, lateral flow near the surface becomes increasingly important, and topographic controls on flow convergence can play a meaningful role.

In this context, we use the SAGA Wetness Index (SWI) as a first-order indicator of the potential for surface and near-surface flow accumulation rather than as a mechanistic predictor of hydrological processes within the peat profile. Similar topography-based indices have been applied in peatland settings to estimate lateral flow paths, soil moisture patterns, or contributing areas, particularly under wet conditions. Examples include applications in restored peatlands (Ikkala et al., 2022), permafrost-affected peatlands (Persson et al., 2012), and riparian and near-stream contributing areas in peatland complexes (Richardson et al., 2012).

We acknowledge that SWI does not capture key peat-specific controls such as hydraulic conductivity contrasts, macroporosity, or vertical preferential flow paths. For this reason, SWI is not used to infer flow mechanisms, but rather to describe the spatial distribution of areas that are more likely to contribute to hydrological connectivity once snow cover disappears. This clarification will be added to the revised manuscript, together with the references.

**Methods**

**RC1C7:** L109: To what is the abbreviation "FMI" pointing to?

**RC1A7:** FMI refers to the Finnish Meteorological Institute, which is the source of the open weather data used. This will be clarified in the revised manuscript.

**RC1C8:** L115: Try to be more precise here. Is this the pH in the surface water or in the soil water?

**RC1A8:** Both in the surface water and pore water, at least in the upper peat layers, based on our unpublished data and other studies in the site (e.g., Järvi-Laturi et al. 2025). This will be clarified in the revised manuscript.

**RC1C9:** Fig. 1: I find it rather unusual to use a copy of a topographic map as a background. All shown features are useless unless explained in a legend. But are all shown features necessary to know? Isolines lack numbers! Potentially use the right map to show the location of the research station/ precipitation sampler.

**RC1A9:** We agree that the map features can be changed to better serve the purpose. The figure will be edited in the revised manuscript by changing the background topographic map only to the most relevant features (contours and peatlands) and including the location of the research station.

**RC1C10:** L177: Explain GCPs.

**RC1A10:** GCPs refer to ground control points. This will be added in the revised manuscript.

**RC1C11:** Fig. S2: This relationship looks rather weak. Can you report on the R2 and bias and have you used the confidence interval later on as shown (but not described) here?

**RC1A11:** We agree with the reviewer that the initial calibration shown in Fig. S2 is associated with considerable uncertainty, and we appreciate the opportunity to clarify and improve this aspect of the manuscript. The online DOC sensor was initially calibrated following the manufacturer's recommendation for local site calibration, using a linear model with zero intercept. Due to the limited number of grab samples available at the time and occasional interference by particulate material during high-flow conditions, the resulting relationship showed substantial scatter, which motivated the reviewer's concern.

For the revised manuscript, we therefore recalibrated the DOC time series using an improved method with an expanded dataset that includes a longer monitoring period and a substantially larger number of grab samples. This revised calibration resulted in a marked improvement in the sensor–laboratory relationship (updated $R^2$ and regression statistics will be reported in Fig. S2), and reduced systematic bias across the observed concentration range.

As a result of the recalibration, the absolute range of DOC concentrations increased (revised range 4.3–11.3 mg L$^{-1}$, previously 4.6–6.5 mg L$^{-1}$), while the temporal dynamics and relative event-scale patterns remained consistent. All analyses in the revised manuscript are performed using the updated DOC dataset. Importantly, the main conclusions of the study rely primarily on relative changes in DOC concentration and DOC–discharge relationships rather than absolute concentration values. We therefore consider the interpretations to be robust despite remaining uncertainties in

sensor-based concentration estimates. The revised manuscript will explicitly acknowledge calibration uncertainty and clarify how it affects, and does not affect, the interpretation.

**Results**

**RC1C12:** I find the description of snowmelt with 5 figures and 2 tables as too extensive. Information can be transferred in a more condensed way focusing on the information that is really needed in the subsequent analysis.

**RC1A12:** We agree that the information can be presented more concisely. This section will be revised by integrating some of the figures and tables and moving some of the figures to supplementary materials.

**RC1C13:** L291: What caused the rapid snowmelt? Temperature? Rainfall on snow?

**RC1A13:** The rapid melting of snow cover was mainly driven by air temperatures (mean 7.9°C between 12 and 18 May as stated in section 3.2, L360). However, there was a small rain event on 16 May, which might have further accelerated the melt.

**RC1C14:** L292: To what statistic measure "variation" is referring to? Can you be more quantitative here?

**RC1A14:** This refers to Figure 3, which shows the snow depth histograms for each survey. Variation refers to the distribution of snow depth in all pixels, which represents the spatial variation of snow depth. Will be clarified in the revised manuscript.

**RC1C15:** L331: For me it is a surprise that the SWI is dynamic. The description in the method does not point to SWI being used as a dynamic surface feature. I interpreted it as a static topography feature. So, for me this is hard to understand. Do you describe snow melt in different classes of the snow-free topography or temporally dynamic SWI of the snow surface?

**RC1A15:** It is correct that SWI is a static topography-based measure, and the dynamic aspect here is the changing snow cover. At L331, mean SWI refers to SWI of the snow-free areas, where the mean SWI is calculated for all pixels where snow depth has reached zero, giving an estimate of the potential wetness of melted areas. Similarly, SWI classes are calculated based on snow-free topography. We will explain this more clearly in the methods in the revised manuscript.

**RC1C16:** L334f: Can you state if difference were statistically significant?

**RC1A16:** Based on pairwise Wilcoxon test, differences in snow depth between SWI classes were statistically significant (p < 0.01), except for snow depth in low and medium classes on 16 May. Similarly, differences in snow depth change between SWI classes were statistically significant, except for low and medium classes on 14–15 May. This information will be added to the revised manuscript.

**RC1C17:** Table 3: For me it would be a better option to show the content of that table in Fig. 6 as a third panel.

**RC1A17:** We thank the reviewer for the good suggestion, which would also help to reduce the number of tables. Information in Table 3 will be added to Fig. 6 in the revised manuscript.

**RC1C18:** Chapter 3.2: The title implies a description of DOC only but the chapter contains much more.

**RC1A18:** Thank you for pointing this out. We agree that the current title does not fully reflect the content of the chapter. The title will be changed to "High frequency hydrological and dissolved organic carbon (DOC) time series" in the revised manuscript.

**RC1C19:** Fig. 7: Use TSSeq concentrations in the axis as well. Is this mg/L as a unit or rather unitless? What is the data source for snow depths here? Is this the same data as described above (UAS)? I am not sure if there is a reason to display load of DOC and TSS concentrations in the same plot. Same for WTD and water temperature.

**RC1A19:** The sensor gives TSS value in mg/L, but without local calibration, the values should be interpreted primarily in terms of their temporal dynamics rather than absolute concentrations. In Fig. 7, TSS is shown on a log scale for the ease of interpretation.

The snow depth shown in Fig. 7 is derived from a point snow depth measurement at the location indicated in Fig. 1. This dataset is shown instead of the UAS-derived snow depth to provide a continuous reference over the longer monitoring period. This distinction will be clarified in the figure caption.

DOC load and TSS are shown together to limit the size and number of panels, however we agree that other options could be considered and will reconsider this in the revised manuscript. WTD and (ground)water temperature are monitored in the same GW well using the same logger, and we therefore consider their joint presentation to be meaningful.

**RC1C20:** L355: Consider a different wording as "followed by" implies that first discharge increased and then air temperature and rainfall increased (that actually triggered the discharge?).

**RC1A20:** Will be changed in the revised manuscript.

**RC1C21:** L362-374: Concentration of DOC hardly change over time so that the discharge dynamics are the overwhelmingly dominant driver of the load, right? This stark differences in the variation of both could be mentioned here.

**RC1A21:** We agree that the changes in DOC concentrations are minor compared to Q and that it is discharge that drives the DOC load, and this remark will be added in the revised manuscript.

**RC1C22:** L398: This statement puzzles me as the relative position of DOC and Q in the plot (Fig. 8) is matter of the scaling of the two different Y-axes.

**RC1A22:** We thank the reviewer for pointing this out. We agree that the relative position of DOC and Q in a dual-axis plot is inherently dependent on the scaling of the two y-axes. The intention of Fig. 8 was to compare the timing and general shape of DOC and discharge responses during the event. However, we acknowledge that the current wording and figure presentation may invite misinterpretation. In the revised manuscript, we will remove or rephrase text that implies inference based on the relative position of the two curves and revise the figure to focus explicitly on temporal co-variation rather than visual comparison of magnitudes.

**RC1C23:** Fig. 8: Why is cumulative Q and SWE loss with the same unit referring to different Y-axes. This should be on the same axis. I have troubles understanding the SWE loss in the figure. Majority happens at the 9[th] May – I see that this is due to a gap in the data. However, the way to display this is not helpful. Consider to leave out the vertical line starting from 0.

**RC1A23:** Cumulative Q and SWE loss are shown on separate y-axes because their magnitudes differ substantially. SWE loss is estimated for the entire study area based on UAS-derived snow depth maps, whereas cumulative Q is measured at the stream gauging station, resulting in much smaller values. We agree, however, that displaying variables with the same units on separate axes can be confusing and complicates interpretation, and thus, Fig. 8 will be edited in the revised manuscript.

SWE loss is calculated based on subsequent UAS surveys, and the gap between the first survey on 1 May and the next on 14 May causes the major increase. The step direction is set to the center to represent the midpoint between the surveys, and this is why the majority of changes seem to happen on 9 May. We agree that the current visualization is not optimal. In the revised manuscript, we will revise Fig. 8 to improve clarity, including reconsidering the step representation and removing the vertical line starting from zero, as suggested.

**RC1C24:** L433-435: Some of the information are redundant here as anticlockwise and HI<0 is the same thing.

**RC1A24:** We thank the reviewer for pointing this out. We agree that describing anticlockwise hysteresis and HI < 0 is redundant, and the text will be revised to remove this redundancy in the revised manuscript.

**RC1C25:** Fig. 9: Use TSSeq on the axis.

**RC1A25:** Will be changed in the revised manuscript.

**Discussion**

**RC1C26:** I have issues with the cut between chapter 4.1, 4.2 and 4.3. For me the separation is not clear but redundancies are large. Discussion circles around the same processes that are explained by different data in the different chapters. The idea of a discussion should be more integrative and less along the steps of the result section, especially when the same processes are discussed.

**RC1A26:** We thank the reviewer for this comment and agree that the current separation between Sections 4.1-4.3 is not sufficiently clear and leads to redundancy. In the revised manuscript, we will reorganize the Discussion to be more explicitly process-oriented and integrative. Instead of structuring the Discussion along individual result sections, we will synthesize snow cover evolution, groundwater dynamics, isotope information, and DOC–Q relationships within a smaller number of conceptually focused subsections. This revised structure will reduce repetition and place greater

emphasis on linking multiple observations to common hydrological and biogeochemical mechanisms.

**RC1C27:** L462: Again, I have issues to make the link of snowmelt and SWI. A high SWI marks areas in the landscape that tends to be wetter as flow paths converge (large upstream area, low slope) while low SWI values mark areas that are steeper and have smaller upstream area. How does that come together with the snow melt? In a direct causative way? Or because both are a function of topography? Steeper hillslopes do not allow for snowpack accumulation and are more exposed to radiation... So how does that link to connectivity in the landscape?

**RC1A27:** We thank the reviewer for raising this important conceptual point. We agree that the SAGA Wetness Index (SWI) does not causally control snow accumulation or snowmelt, and that both snow distribution and SWI are influenced by topography and associated factors such as slope, contributing area, and radiation exposure. We do not intend to imply a direct causal link between SWI and snowmelt processes. In this study, SWI is not used to explain where or why snow melts, but rather to characterize the hydrological relevance of areas once they become snow-free. Snowmelt acts as a temporal trigger that progressively activates different parts of the landscape, whereas SWI describes the potential for lateral flow convergence and hydrological connectivity in those activated areas.

Within this framework, snowmelt occurring in low-SWI areas is less likely to result in rapid stream connectivity due to limited flow convergence or greater infiltration potential, whereas snowmelt in high-SWI areas is more likely to enhance connectivity and contribute disproportionately to discharge and solute transport. The observed co-variation between snow-free area weighted by SWI and event-scale hydrological and biogeochemical responses therefore reflects an interaction between static topographic controls and the dynamic progression of snowmelt, rather than a direct control of SWI on snowmelt itself. We will revise the manuscript text to clarify this distinction and avoid any implication of causality between SWI and snowmelt

**RC1C28:** L478f: The ice cover is a new result brought up here. For me it does not really explain why ice is forming here especially.

**RC1A28:** We agree that this interpretation can be stated more clearly. The low-lying areas refer to microtopographical depressions where the water level is above the ground surface, which is easily frozen during the winter. The depressions can also capture more snow due to wind trapping, further increasing the persistence of the ice cover. This process will be clarified in the revised manuscript.

**RC1C29:** L503: This relative increase was not convincingly shown nor quantified in the result. So, it is a bit hard to follow that argument here.

**RC1A29:** We thank the reviewer for this comment and agree that the wording of this argument is ambiguous. The statement refers to changes in the C-Q dynamics rather than to concentration increases alone, specifically the lack of clear dilution as snow-free areas expanded, particularly in high-wetness zones. In the revised manuscript, we will clarify this interpretation and explicitly link it to the relevant results (e.g., C-Q behavior and hysteresis patterns) or revise the wording to avoid implying a quantified increase where this is not directly shown.

**RC1C30:** L509f: However, consider that the dilution effect is very small with concentration hardly changing during the event. This speaks rather for a transport and not a source limitation of DOC. So, from my observation I see very mild dilution effects only and therefore nearly every flow path loaded with DOC and therefore no major changes in sources of flowpath. This is basically a chemostatic system.

**RC1A30:** We thank the reviewer for this insightful interpretation and agree that the observed DOC dynamics are consistent with near-chemostatic behavior and predominantly transport-limited DOC export. DOC concentrations vary only modestly during the snowmelt events compared to the large changes in discharge, indicating that most mobilized flow paths are DOC-rich and that dilution effects are generally weak.

We agree that this behavior does not support major shifts in DOC sources or flow paths, but rather suggests that event-scale dynamics primarily reflect changes in transport efficiency and hydrological connectivity within an otherwise well-buffered system. Hysteresis direction and flushing indices therefore indicate subtle differences in timing and routing of DOC transport, rather than fundamental changes in source contributions. Despite the subtle changes, understanding these processes has great importance in understanding the DOC transport mechanisms from peatlands to downstream water bodies on a wider catchment scale.

In the revised manuscript, we will strengthen this interpretation by explicitly framing the system as near-chemostatic, reducing language that implies strong dilution or rapid source depletion, and clarifying that DOC–Q dynamics mainly reflect transport-related processes during snowmelt.

**RC1C31:** L538-553: For me this discussion repeats former statement but add some TSS data. I suggest to strongly reduce redundancies and combine with the discussion above.

**RC1A31:** We agree that the organization of the discussion section can be improved. Discussion will be revised for clearer separation for the revised manuscript.

**RC1C32:** L555ff: This is a long statement for a rather simple fact. Nearly invariant concentrations multiplied with highly variant discharge will result in a load that is exactly the same as the discharge.

**RC1A32:** We agree that the statement can be formed more concisely to express the point. Will be edited in the revised manuscript.

**RC1C33:** L592ff: Again, I would be careful in interpreting the mild concentration changes too much. Yes, the described processes are meaningful but I don't think we see a fundamental change of flow paths and sources but rather slight changes. So, phrases such as "quick depletion" or "sudden depletion" are a bit too much for me.

**RC1A33:** We thank the reviewer for pointing this out and agree that the wording can be improved. With 'quick' and 'sudden', we refer to shifts in behavior in a short (daily) timescale, rather than absolute concentration values. This will be stated more clearly in the revised manuscript. For the revised manuscript, we will strengthen the discussion regarding the variation of concentrations (or lack of it) to avoid over-interpretation. As pointed out by the reviewer, the processes and dynamics are meaningful, but the concentration changes can be better discussed.

---

## Author Comment (AC2)

**RC2C1:** This study presents the use of Unmanned Aircraft System (UAS) snow cover/depth measurements on a fen to complement the interpretation of how the dynamics of source areas and flow paths influence the export of dissolve organic carbon (DOC) from a small, high latitude catchment during snowmelt. Oxygen isotopes, continuous groundwater level measurements and snow surveys, as well as continuous measurement of an optical proxy for DOC are valuable complements to the data used in the analysis. Hysteresis and flushing indices are a key part of the reasoning used in ascribing the role of hydrological connectivity bringing in new source areas and dilution of source areas during events.

**Major concerns:**

**RC2C2:** Much emphasis is placed on interpreting the variation in DOC concentrations. However what strikes me more is the overall stability of DOC when the authors say that as much as 50% of the runoff peaks could come from snowmelt/precipitation inputs that are likely to be sources with little DOC.

**RC2A2:** We appreciate this observation and agree that the relative stability of DOC concentrations, despite substantial inputs of low-DOC snowmelt/precipitation during events, is a central result of our study. In the revision we will make this point explicit and frame the system as exhibiting near-chemostatic, predominantly transport-limited behavior: DOC concentrations vary modestly compared to discharge, so DOC load is largely governed by Q rather than by large concentration changes.

To align the interpretation with this behavior, we will (i) emphasize in the Results that event-scale DOC variability is small relative to discharge variability, (ii) discuss hysteresis direction and flushing indices as indicators of subtle timing/routing differences in transport rather than evidence of pronounced change in sources, and (iii) integrate this perspective in the Discussion as our primary interpretive frame during snowmelt.

In addition, we will include an isotope-based hydrograph separation for the peak melt period to quantify the contribution of recent snowmelt to streamflow. This analysis will allow us to state more precisely that the observed stability of DOC occurs even when event water fractions are elevated, reinforcing a transport-limited interpretation. Together, these revisions will clarify that the value added by the UAS observations lies in constraining where and when the landscape becomes hydrologically active, while the DOC response remains comparatively buffered.

**RC2C3:** The interpretation of the rich data sources in this paper would be much more convincing if the different catchment DOC sources and pathways alluded to in the

discussion were quantified in an approach that keeps track of mass balances of water and carbon. Without that, I am concerned that conclusions may be drawn that are at odds with what the data show. In such a quantification, care should also be taken to explain what the inclusion of the UAS data adds to the interpretation of the other data.

**RC2A3:** We thank the reviewer for this important comment and agree that a coupled water–carbon mass balance would provide the most robust framework for quantifying DOC sources and pathways. We acknowledge that, with the available data, it is not possible to close a catchment-scale carbon mass balance with sufficient confidence, and we do not aim to do so in this study.

In the revised manuscript, we therefore more clearly delimit the scope of our interpretations to event-scale DOC–Q dynamics and relative changes in hydrological connectivity, rather than absolute source contributions or carbon budgets. Stable water isotopes allow us to quantify the relative contribution of recent snowmelt to runoff through an explicit hydrograph separation, which will be added in the revised manuscript. This strengthens the quantitative basis for interpreting the proportion of low-DOC event water contributing to discharge during peak melt.

The added value of the UAS-derived snow depth data lies not in constraining water or carbon mass balances directly, but in resolving the spatial and temporal heterogeneity of snow accumulation and melt across the fen. This information constrains where and when hydrologically relevant areas become activated during snowmelt, thereby improving interpretation of runoff generation, connectivity, and the timing of low-DOC inputs to the stream. In the revised manuscript, we will clarify this role of the UAS data and avoid interpretations that would require explicit carbon mass balance closure.

**Minor comments:**

**RC2C4:** Section 2.1. Is there any information on how well this groundwater level represents the spatial variability in GW level across the fen?

**RC2A4:** We had 3 similar GW wells on the site during the study, and the GW level dynamics were very similar in all of those, although the GW levels varied by some centimeters. We chose this particular well because it is located closest to the stream gauging station and was the first to thaw, meaning that it more likely represents the first areas where the GW level reaches below the ground surface.

**RC2C5:** Is there information on how the water from the larger, upslope catchment area moves through the fen? Are subsurface, preferential flowpaths an important feature?

**RC2A5:** We thank the reviewer for pointing out this important concern. We expect that the SAGA Wetness index provides a reasonable estimate of the main surficial flow paths through the fen, as it is calculated based on the whole upslope catchment area, and only the map extracted for the study site is shown in the manuscript.

Based on the hydraulic conductivity measurements done at the site, there is a layer with higher conductivity at approximately 30-50 cm depth, which can act as a preferential flow path. However, surficial flow paths likely dominate, especially during the peak melt period, due to the quick melting of snow and soil frost. We also took pore water samples from the peat profile during the study period, but the function of the preferential flow path could not be clearly identified from those. We will include the discussion of this potential preferential flow path in the revised manuscript.

**RC2C6:** Section 2.2 – Were the daily stream isotope samples taken at the same time each day. During spring, the runoff can have strong diurnal variation, so where one samples in that diurnal variation is important.

**RC2A6:** This is an important concern which we acknowledge. During the surveys on the peak melt period, samples were collected with a maximum 2 hours difference. However, most of the isotope samples were collected by the research station staff and taken within a few hours' difference, although, unfortunately, the exact sampling time cannot be confirmed.

**RC2C7:** Section 2.3 – While there is good coverage of snow depth in the open fen, is there any information about the progress of snowmelt in the larger, upslope forested area which is presumably a large source of the water measured at the flume?

**RC2A7:** Due to challenges of applying the UAS survey method to larger and forested areas, we restricted this study to the open peatland area and thus have no data from the snowmelt in the upper catchment. However, the research station has a forest measurement site with point snow depth monitoring located approximately 600 m downhill from the Puukkosuo study site. Average snow depth in this forest site was 28 cm on 14 May and 12 cm on 17 May. Corresponding snow depths could be assumed in the forested upslope catchment of the study area. This information will be included in the revised manuscript.

**RC2C8:** How well does the 10 cm resolution in elevation measurement of each cell compare to the variation of the topography across the fen? Is this important for assessing how well TWI can be identified.

**RC2A8:** SAGA wetness index was calculated on 50 cm DEM, and snow depth maps were resampled to a 50 cm resolution for the SWI-snow depth comparisons. The 50 cm resolution for SWI calculation was selected because it was fine enough to capture the influence of microtopography. Also, SAGA algorithm has been successfully used to represent soil moisture at fine resolutions (≤2 m, Riihimäki et al. 2021). Based on visual assessment and observations on site, 50 cm resolution provided the most accurate representation of flow paths. Finer resolutions are typically unsuitable for topographical analysis, and coarser resolutions did not capture the smaller flow paths (which have been observed on site) correctly.

**RC2C9:** Snow water equivalent was apparently measured at 5 points. What is the variation between points? This should be of relevance for the assumption of a uniform SWE across the fen. While an average SWE may be relevant for the water balance of the catchment, if one wants to consider local inputs of water to the catchment, then that variation in SWE will be important. A relationship between degree of change in snow depth and SWE from your point measurements might be a way to improve precision on where water is being input to the fen surface. The use of a reference from alpine areas (Lopez-Moreno) to argue that SWE is relatively uniform may be problematic. It would be better to know how SWE varies in a high-latitude fen as opposed to an alpine area with much more relief and different diurnal patterns of insolation.

**RC2A9:** We thank the reviewer for this insightful comment. SWE was measured at a single point, which will be clarified in the manuscript, and the measurement location will be added to the study site map (Fig. 1) for clarity. Results of the manual SWE measurement are reported in Table 2.

In calculating SWE for each cell of the study area, we used the snow depth retrieved from UAS data and a one-point measurement of snow density and thus assumed snow density to be uniform. Although we do acknowledge the limitations of one-point measurement of snow density, the study site is rather homogeneous in terms of vegetation cover and topography, and thus, minor variation in snow density can be expected. Topography and especially slope likely have some impact, but again, we expect this impact to be higher on snow depth than density, as shown by e.g. López-Moreno et al. (2013). Similarly, the study of Whittington et al. (2012) in James Bay Lowlands, considering different peatland types, also showed that no significant differences in snow density across landscape types were found. The reference will be added to the revised manuscript.

**RC2C10:** Section 2.4.1 – is there any test of how well the processing of UAS images was able to capture the variation of snow depth and SWI across the fen?

**RC2A10:** The processing method for UAS images aimed for the best model accuracy (XY and Z accuracy), which are presented in Table 1. We calculated the error metrics for each survey by comparing UAS snow depth to the manual snow depth measurements done at 7 points on the snow transect. This resulted in a mean RMSE of 9.79 cm for all surveys. Error metrics for each survey can be found in Table S1. SWI was calculated from LiDAR data, and no ground truth data was available for accuracy assessment.

**RC2C11:** Line 251: Data is plural, so "data was" should read "data were".

**RC2A11:** Will be corrected in the revised manuscript.

**RC2C12:** Line 270-280 –MAJOR It appears that different source areas are being assumed AND that the concentration of the source areas is changing. How can one distinguish between changes in source area concentration and changes in source areas? Please be more clear about the assumptions being used in the hysteresis and flushing analyses.

**RC2A12:** We thank the reviewer for this important comment. Hysteretic behavior can be used to associate with different source areas and/or flow paths, while flushing behavior is associated with transport/source limitation of the solute. Direct quantification of source-area concentrations is not possible with the available data and hysteresis analysis, instead, inferences are made based on the relationship between concentration and discharge. In this study, hysteresis patterns are used to characterize the relative timing of changes in concentration with respect to discharge during individual events, which may reflect differences in flow routing, connectivity, or transport pathways, but do not uniquely identify specific source areas or their concentrations. Similarly, the flushing index is used to indicate whether solute export during an event appears transport-limited or supply-limited, but it does not provide direct information on absolute source concentrations.

In the revised manuscript, we will clarify these assumptions and explicitly distinguish between what is inferred from hysteresis and flushing patterns and what cannot be resolved from the data.

**RC2C13:** Section 3.1, Line 299 – should RMSE have the unit of cm?

**RC2A13:** Yes, RMSE is reported in centimeters, consistent with the units of snow depth.

**RC2C14:** Fig. 3. Please include uncertainty in this. It is worry ing that such a large portion of the predicted data is excluded as unreasonable. What does the non-excluded data actually say about the snow depth, especially when the ambition is to map the spatial variability of the snow depth.

**RC2A14:** We estimated the accuracy of UAS snow depth maps by calculating the error metrics by comparing UAS snow depth to manual snow measurements (Table S1). Negative values are more likely to represent shallow or zero snow depth within the measurement uncertainty. Thus, we decided to exclude them in Fig. 3 to ease the interpretation of snow depth, particularly in cells that still confidentially have snow cover. Uncertainty will be visualized in the revised manuscript by adding RMSE uncertainty bounds in Fig 3.

**RC2C15:** Lines 330-335. The relation of SWI and snowdepth is discussed. Does the SWI get influences by snowdepth?. I presume SWI is calculated during the snow-free period, but it would be good to clarify this in the methods.

**RC2A15:** We agree that the relation of SWI and snow depth can be explained more clearly in the manuscript. The SWI index is calculated from LiDAR data collected during snow-free period. SWI is invariable, but we discuss the mean SWI of snow-free areas, which are progressing through the study period. In this study, we do not intend to imply a direct causal link between SWI and snow depth but use SWI to describe the potential hydrological connectivity of these areas as they become snow-free. We will clarify this rationale in the methods section in the revised manuscript.

**RC2C16:** Line 384-388 The change in groundwater levels from being above the soil surface until the final runoff peak, after which the water table is below the soil surface seems to indicated a very important shift in flow paths and hydrological connectivity in the fen. Yet this major change gets little mention and does not seem to have a major impact on the DOC concentration. This seems to deserve more attention in the discussion where groundwater does not get much mention.

**RC2A16:** We thank the reviewer for pointing this out and agree that the position of GW level has significant implications for flow paths and hydrological connectivity, and this aspect could be strengthened in the discussion. The GW level remains less than 5 cm below the surface until the end of the study period, and these surficial layers of peat can be efficiently flushed during the snowmelt or can potentially have limited DOC due to the freeze-out effect, which could explain the minor impact on DOC concentration. We will improve the discussion of GW level by adding this interpretation in the revised manuscript.

**RC2C17:** Figure 7 – The "snow" value of the oxygen isotopes:  is that sampled in the snow pack I assume? Please make that clear. If it is the isotope signature remaining in the snowpack, this means the snowpack is contributing meltwater with a less depleted isotope ratio. This complicates the interpretation. A quantitative hydrograph separation would be a much more satisfactory basis for the interpretation of the runoff sources. I would recommend an explicit hydrograph separation. There will of course need to be assumptions, but then the assumptions will be explicit.

**RC2A17:** We thank the reviewer for this important concern and suggestions. Yes, the snow value is measured from snowpack, sampled as described in methods: "During the field campaigns, snowpack samples were collected for stable water isotope analysis. The sample was collected by coring the whole snow profile from the top to the base of the snowpack with a snow tube corer (diameter 3.5 cm)". For clarity, the "snow" will be renamed as "snowpack" in the legend in the revised manuscript.

We agree with the reviewer that the isotope signature of the snowpack and associated fractionation during melt complicate the interpretation of runoff sources. In the revised manuscript, we will therefore add isotope-based hydrograph separation for the peak melt period to provide a more quantitative assessment of runoff source contributions. The underlying assumptions of hydrograph separation will be clearly stated and discussed.

**Results**:

**RC2C18:** Much is made about the variation in the DOC concentration, but what strikes me is how stable the DOC is, especially when the precipitation and snowmelt entering the catchment on top of frozen fens soils are potential sources of runoff with almost no DOC until they interact with the subsurface. But that interpretation depends on quantifying the inputs of "recent precipitation" along flow paths that stay out of the soil/peat. The fact that the groundwater well is frozen, with water tables above the peat surface during most of the study period (Fig. 7e) would indicate that the input of precipitation/snowmelt on the mire that makes it to the stream would be a source without much DOC at all.

**RC2A18:** We thank the reviewer for this insightful comment. We agree that, despite event-scale dynamics, DOC concentrations remain relatively stable throughout much of the study period, even when a large fraction of snowmelt inputs with low DOC are expected to contribute to runoff. This suggests that a fraction of streamflow continues to interact with DOC-rich zones, rather than being dominated by flow paths that bypass the soil or peat entirely.

In the revised manuscript, we clarify this interpretation and explicitly acknowledge that stronger dilution might be expected under dominant overland flow conditions on frozen peat. The persistence of relatively stable DOC concentrations therefore implies that subsurface interaction is occurring, likely due to spatially and temporally heterogeneous thaw conditions and flow routing. As indicated by the snow depth maps, snow cover, and by inference soil frost, is not spatially uniform. For example, in the immediate adjacency of the spring, no soil frost was observed, allowing infiltrating water to access peat layers and acquire DOC even during periods when frozen conditions dominate elsewhere in the fen.

Additionally, for the revised manuscript, we decided to recalibrate the DOC data, as a higher number of grab samples and longer time series significantly improved the calibration. This will lead to a slightly increased deviation in DOC concentrations (4.3–11.3 mg L, when previously 4.6–6.47 mg L), but the overall dynamics remain consistent. We will also improve the interpretation by considering the variation (or lack of it) in DOC concentration during the events.

**RC2C19:** Figure 8 – please find a way to help the reader better compare panes (a) and (b) of the figure. Stacking one on top of the other, rather than side by side would be one easy way, even if it would take more space.

**RC2A19:** We thank the reviewer for these suggestions. We agree that this figure can be improved for better readability, and it will be edited in the revised manuscript.

**RC2C20:** Line 480. The role of ice is mentioned, with the UAS snow depth analysis interpreting surficial ice as snow. Please say more about "formation of ice in low lying areas". My experience is that the upper centimetres of mires are often frozen during the winter, whereas forested upland areas often do not have much ice in the soil profile. Is the entire fen frozen to a certain depth (some centimetres), or is it a crust of ice on the snow and or soil surface that is referred to here. This is good to clarify since the depth of soil frost has a bearing on the interpretation of flow paths and where the DOC sources are that keep DOC from diluting more as snow melts on the fen, creating large inputs of low-DOC water close to the flume.

**RC2A20:** The ice layer here refers to the ice cover formed in wet depressions, where the water table is above the peat surface and easily freezes, forming a thick ice cover. This will be explained more clearly in the revised manuscript. Soil frost is likely present across much of the fen but is spatially heterogeneous. Unfortunately, we did not have data on the depth of soil frost, but this is supported by the field observations, where in

some locations, soil frost was not present (a metallic stick could be pressed into the peat), and in some peat was frozen throughout the study period.

This is an important concern and refers to the central insight of the study: snowmelt and soil thaw are not spatially uniform across the peatland, leading to dynamic generation of flow paths and influencing stream DOC response.

**RC2C21:** Speaking of the flume, does it capture all the water running off from the upslope areas? With water tables above the ground surface on a relatively flat landscape, there might be a possibility of water bypassing the flume.

**RC2A21:** The flume location ensures that it captures only water that is passing through the peatland, but some water bypassing is expected. Based on the flow accumulation analysis and wetness index, a total of three outlets can be identified in the fen. We assume that the bypassing water is not significantly different in terms of volume or water quality, and thus does not impact on our analysis results. However, we acknowledge that this is an assumption and the possible limitations of it. We will include mentioning this source of uncertainty in the discussion of the revised manuscript.